# α-Bisabolol Attenuates NF-κB/MAPK Signaling Activation and ER-Stress-Mediated Apoptosis by Invoking Nrf2-Mediated Antioxidant Defense Systems against Doxorubicin-Induced Testicular Toxicity in Rats

**DOI:** 10.3390/nu14214648

**Published:** 2022-11-03

**Authors:** Seenipandi Arunachalam, Mohamed Fizur Nagoor Meeran, Sheikh Azimullah, Niraj Kumar Jha, Dhanya Saraswathiamma, Alia Albawardi, Rami Beiram, Shreesh Ojha

**Affiliations:** 1Department of Pharmacology and Therapeutics, College of Medicine and Health Sciences, United Arab Emirates University, Al Ain P.O. Box 15551, United Arab Emirates; 2Department of Biotechnology, School of Engineering and Technology (SET), Sharda University, Greater Noida 201310, Uttar Pradesh, India; 3Department of Pathology, College of Medicine and Health Sciences, United Arab Emirates University, Al Ain P.O. Box 15551, United Arab Emirates; 4Zayed Bin Sultan Center for Health Sciences, College of Medicine and Health Sciences, United Arab Emirates University, Al Ain P.O. Box 15551, United Arab Emirates

**Keywords:** α-bisabolol, doxorubicin, testicular dysfunction, NF-κB/MAPK signaling, ER stress, apoptosis

## Abstract

The present study investigated the effects of α-bisabolol on DOX-induced testicular damage in rats. Testicular damage was induced in rats by injecting DOX (12.5 mg/kg, i.p., single dose) into rats. α-Bisabolol (25 mg/kg, i.p.) was administered to the rats along with DOX pre- and co-treatment daily for a period of 5 days. DOX-injected rats showed a decrease in absolute testicular weight and relative testicular weight ratio along with concomitant changes in the levels/expression levels of oxidative stress markers and Nrf2 expression levels in the testis. DOX injection also triggered the activation of NF-κB/MAPK signaling and increased levels/expression levels of pro-inflammatory cytokines (TNF-α, IL-6, and IL-1β) and inflammatory mediators (iNOS and COX-2) in the testis. DOX triggered apoptosis, manifested by an increment in the expression levels of pro-apoptotic markers (Bax, Bcl2, cleaved caspase-3 and -9, and cytochrome-C) and a decline in the expression levels of anti-apoptotic markers (Bcl-xL and Bcl2) in the testis. Additionally, light microscopy revealed the changes in testicular architecture. α-Bisabolol rescued alterations in the testicular weight; restored all biochemical markers; modulated the expression levels of Nrf2-mediated antioxidant responses, NF-κB/MAPK signaling, endoplasmic reticulum (ER) stress, and apoptosis markers in DOX-injected testicular toxicity in rats. Based on our findings, it can be concluded that α-bisabolol has the potential to attenuate DOX-induced testicular injury by modifying NF-κB/MAPK signaling and the ER-stress-mediated mitochondrial pathway of apoptosis by invoking Nrf2-dependent antioxidant defense systems in rats. Based on the findings of the present study, α-bisabolol could be suggested for use as an agent or adjuvant with chemotherapeutic drugs to attenuate their deleterious effects of DOX on many organs including the testis. However, further regulatory toxicology and preclinical studies are necessary before making recommendations in clinical tests.

## 1. Introduction

Cancer treatment using chemotherapeutic agents substantially compromises physiological and biochemical homeostasis and triggers multiple organ failure in the course of therapeutic procedures [1]. Many chemotherapeutic agents are gonadotoxic, leading to infertility, and this fertility potential had a huge impact on the quality of life according to cancer survivors [2]. Among numerous chemotherapeutic agents, doxorubicin (DOX), a predominant anti-neoplastic agent, has acquired immense popularity over a few decades due to its usefulness in the management of various hematological and solid tumors. However, its clinical use has been restricted owing to its lethal multiple organ toxicities [3,4,5,6]. The testis is among the major non-target organs which are highly vulnerable to the lethal side effects of DOX; these effects strikingly impede the process of spermatogenesis and eventually lead to infertility [7]. DOX-chemotherapy-associated testicular damage involves an oxidative injury that is mediated by the formation of oxyradical complexes containing hydroxyl and superoxide radicals and traces of iron [8]. DOX is a well-known cytotoxic agent functioning through topoisomerase II inhibition and intercalation of DNA in rapidly developing malignant cancer cells [9]. Even though the exact underlying mechanism of DOX-induced testicular toxicity is still not fully understood, published evidence revealed that DOX-induced testicular toxicity primarily involves the combination of various pathophysiological events including oxidative stress, lipid peroxidation, inflammation, and cellular apoptosis [10].

Inflammation is a potential counteraction against pathogens, injured cells, and toxic chemicals by the immune system in which the levels of inflammatory cytokines and mediators are altered by the macrophages [11,12]. In recent years, numerous studies revealed a strong interlink between DOX assault and the activation of nuclear factor kappa B (NF-κB), which participates in the regulation of genes that encode inflammatory cytokines and apoptotic cell death [13]. In addition, the inflammatory cytokines are also regulated by the mitogen-activated protein kinase (MAPK) signaling, including p38 MAPK [14]. To modulate inflammatory cascades, p38 MAPK also participates in controlling cell cycle and death. DOX triggered p38 MAPK activation and its role in promoting apoptosis is well documented from earlier reports [15].

Apart from the involvement of mitochondrial apoptotic pathways, recently, endoplasmic reticulum (ER) stress has also been recognized as playing a vital role in the organ injuries caused by DOX. In addition to the mitochondrial apoptotic pathways, targeting ER stress received enormous attention for protective maneuvers in DOX-associated organ injuries. Increased ROS production has a bidirectional relationship with the ER, and oxidative stress results in unfolded protein accumulation in the ER [16]. Upon initiation of ER stress, GRP78, a key ER stress sensor, is released and facilitates programmed cell death [17].

Since oxidative stress plays a major role in DOX-induced testicular injury, the protective mechanism might be dependent on invoking antioxidant defense mechanisms, antioxidants (SOD, catalase, Gpx), and nuclear factor erythroid factor 2-related factor 2 (Nrf2) signaling against oxidative stress. Nrf2 is a master oxidative stress regulator and a chief regulator in orchestrating redox defense mechanisms during stress conditions via activation of HO-1 and antioxidant defense systems [18,19,20]. Additionally, NF-κB was revealed to inversely regulate Nrf2 transcription and activities [19]. Nrf2 activation has been reported to rescue the ER from oxidative stress by scavenging ROS overload [21]. Based on the above-mentioned concepts, numerous antioxidants and anti-inflammatory or antiapoptotic agents were tested to counter DOX-triggered testicular toxicity in rats. Currently, there is no single agent proven efficient enough to reverse or attenuate this life-threatening adverse effect in cancer survivors.

In recent years, a number of studies demonstrated that medicinal plants and phytochemicals derived from plants play an important role in improving male reproductive function markers in healthy and infertile individuals. Among them, numerous phytochemicals have been investigated in DOX-induced testicular injury and shown to reduce oxidative stress and inflammation. In different classes of phytochemicals, sesquiterpenes are recognized as the most active constituents abundantly present in various traditional medicinal plants with plenty of pharmacological properties [22]. α-Bisabolol, a major monocyclic sesquiterpene abundantly present in chamomile (*Chamomilla recutita* L.) [23], the wood of candeia (*Eremanthus erythropappus*) [24], *Plinia cerrocampanensi* [25], and Salvia (*Salvia runcinata*) [22].

Recently, the hydroalcoholic extract of *Matricaria chamomile* had shown protective effects in testicular ischemia–reperfusion in a rat model of torsion/detorsion-induced testis tissue damage [26]. This protective effect was attributed to the presence of α-bisabolol and its antioxidant capacity [26]. α-Bisabolol is one of the major ingredients in dermatological and cosmetic formulations, including lipsticks, baby care products, hand and body lotions, after-sun products, aftershave creams, deodorants, and sun care sports creams. Oral administration of α-bisabolol has been reported to be safe and non-toxic in rats and mice (LD_50_ 13,000 to 14,000 mg/kg body weight) [27]. In addition, α-bisabolol is a very well documented phytochemical in countering lipid peroxidation, oxidative stress, inflammatory signaling cascades, inflammasome activation, mitochondrial dysfunction and apoptosis in isoproterenol (ISO)-induced myocardial ischemia in rats [28,29,30]. However, there is no scientific evidence on the possible protective effect of α-bisabolol on DOX-induced testicular toxicity in rats. In this study, we hypothesized that α-bisabolol may attenuate the testicular toxicity triggered by DOX, and we also investigated the involved molecular mechanisms behind its counteraction against DOX-induced infertility in rats.

## 2. Materials and Methods

### 2.1. Drugs, Chemicals, and Antibodies

α-Bisabolol and DOX were purchased from Sigma-Aldrich (St. Louis, MO, USA). Primary antibodies used for immunoblotting analysis were purchased from Abcam (Cambridge, MA, USA), Santa Cruz Biotechnology (Dallas, TX, USA), and Cell Signaling Technology (Danvers, MA, USA). Secondary biotinylated and horseradish peroxidase (HRP)-conjugated antibodies (goat anti-rabbit/goat anti-mouse) were obtained from Cell Signaling Technology (Danvers, MA, USA). All other chemicals used in this study were of analytical grade.

### 2.2. Experimental Animals

Male albino Wistar rats (220–250 g) were acclimatized for two weeks before the beginning of the experiments in the Animal House of the College of Medicine and Health Sciences (CMHS), United Arab Emirates University (UAEU). The animals were kept in polypropylene cages in a group of four rats at the standard animal house conditions of photoperiod with free access to a chow diet and purified water ad libitum. The experimental procedures were conducted following approval from the Animal Ethics Committee of the UAEU, Al Ain, Abu Dhabi, United Arab Emirates (UAE).

### 2.3. Experimental Design

The animals were randomly divided into four experimental groups, each containing 15 rats. α-Bisabolol was diluted in scientific-grade light olive oil (vehicle), and the solutions were freshly prepared just before administration. A single intraperitoneal injection of DOX (12.5 mg/kg body weight) was administered to the rats to induce testicular toxicity. Group I: normal control rats; Group II: rats treated with α-bisabolol (25 mg/kg, intraperitoneally) daily for a period of 5 days; Group III: rats intraperitoneally injected with a single dose of DOX (12.5 mg/kg) to induce testicular toxicity; Group IV: rats administered a single intraperitoneal dose of DOX (12.5 mg/kg) and α-bisabolol (25 mg/kg, intraperitoneally) for five days. After the treatment duration (i.e., on the 6th day), the rats were anesthetized using pentobarbital sodium (60 mg/kg, body weight) and then sacrificed by cervical decapitation. The isolated testis tissues were snap-frozen in liquid nitrogen for the biochemical and immunoblotting experiments. The testicular tissues were also fixed in the 10% neutral buffered formalin for histological studies. Experimental groups and study design is presented in Figure 1. 

### 2.4. Biochemical Parameters

#### 2.4.1. Estimation of Oxidative Stress Markers

Malondialdehyde (MDA) levels were measured using a commercial detection kit (Northwest Life Science, Vancouver, WA, USA). The activities of SOD and catalase and the concentration of GSH were measured using commercial kits acquired from Sigma Chemicals (St. Louis, MO, USA) and Cayman Chemical Company (Ann Arbor, MI, USA), according to the manufacturer’s instructions.

#### 2.4.2. Estimation of Pro-Inflammatory Cytokines

The levels of interleukin-1β (IL-1 β), tumor necrosis factor-α (TNF-α), and interleukin-6 (IL-6) were measured using enzyme-linked immunosorbent assay (ELISA) kits procured from BioSource International (Camarillo, CA, USA).

### 2.5. Western Blot Analysis

Testicular protein extracts were produced by homogenizing testis samples in ice-cold radioimmunoprecipitation assay buffer mixed with 1X phosphatase and protease inhibitor cocktail (Millipore, Burlington, MA, USA). The homogenate was centrifuged at 1648× *g* for 30 min at a temperature of 4 °C. Samples for immunoblotting were prepared by mixing the supernatant with 4X Laemmli buffer (Bio-Rad, Hercules, CA, USA) and 2-mercaptoethanol (Sigma Aldrich, St. Louis, MO, USA). Equal amounts of protein samples were separated by SDS-PAGE and transferred onto polyvinylidene difluoride membranes (Amersham Hybond P 0.45, GE Health Care Life Sciences, Munich, Germany).

The membranes were incubated at 4 °C overnight with primary antibodies against inducible nitric oxide synthase (iNOS) (1:1000) (anti-rabbit; Sigma Aldrich, St. Louis, MO, USA), nuclear factor erythroid factor 2-related factor 2 (Nrf2) (1:2000), cyclooxygenase-2 (COX-2) (1:500), Bcl2 associated X protein (Bax) (1:500), B-cell lymphoma 2 (Bcl2) (1:1000), p-NF-κB-P65 (1:500), t-IκBα (1:2000), p-IκBα (1:500), P38 (1:1000), p-P38 (1:1000), Bcl-xL (B-cell lymphoma-extra-large), procaspase-3 (1:1000), procaspase-9 (1:1000), cytochrome-C (1:3000) (anti-rabbit and mouse; Abcam, Cambridge, MA, USA), NADPH oxidase-2 (NOX2) (1:1000), NADPH4 (NOX4) (1:1000), glutathione peroxidase-1 (GPX1) (1:2000) (Invitrogen, USA), superoxide dismutase-1 (SOD1) (1:2000), superoxide dismutase-2 (SOD2) (1:2000), catalase (1:2000), tumor necrosis factor-α (TNF-α) (1:1000), interleukin-6 (IL-6) (1:1000), interleukin-1β (IL-1β) (1:500), (Santacruz, Dallas, TX, USA), and cleaved caspase-3 (1:500), and glyceraldehyde-3-phosphate dehydrogenase (GAPDH) (1:5000) (Cell Signaling Technology, Danvers, MA, USA) was employed as a loading control. Membranes were further incubated with their corresponding secondary antibodies (anti-mouse/rabbit) for 1 h at room temperature, and the proteins bands were visualized by using an enhanced chemiluminescence developing kit procured from Thermo Fisher Scientific (Rockford, IL, USA). The signal intensity (densitometry) of the bands was quantified using ImageJ software (NIH, Bethesda, MD, USA).

### 2.6. Estimation of Protein Content in the Testis

The concentration of protein in the homogenate of testis was analyzed using a commercially available Pierce BCA protein assay kit procured from Thermo Fisher Scientific (Rockford, IL, USA).

### 2.7. Histopathological Evaluation

After fixing testicular tissue in neutral buffered formalin (10% *w/v*) for one week, the tissues were gradually dehydrated using different concentrations of ethanol, cleared of alcohol residue in xylene, and lastly embedded in paraffin blocks. The tissues were serially sectioned (5–10 μm) using a microtome (RM2125 RTS, Leica Biosystems, Nussloch, Germany). The sections were stained with hematoxylin and eosin. The tissue sections were mounted on the slides and examined under a light microscope (BX41, Olympus, Tokyo, Japan) using an objective lens of 10× magnification.

### 2.8. Statistical Analysis

The data were statistically analyzed by one-way analysis of variance accompanied by Duncan’s multiple range test (DMRT) using Statistical Package for the Social Sciences (SPSS) software version 25 (IBM, Armonk, NY, USA). The results are expressed as the mean ± standard error of the mean (SEM) for eight rats in each group. The criteria for differences between each group were considered significant at *p* < 0.05.

## 3. Results

### 3.1. α-Bisabolol Prevented Testicular Weight Loss and Oxidative Stress in DOX-Induced Testicular Injury in Rats

DOX-injected rats displayed a significant (*p* < 0.05) reduction in the absolute testicular weight and relative testicular weight ratio; a significant (*p* < 0.05) rise in the level of MDA, a lipid peroxidation product, in the testis; and a significant (*p* < 0.05) reduction in the activities/concentration of testicular SOD, catalase, and GSH. Rats treated with α-bisabolol showed an improved absolute weight of testis and relative testicular weight ratio; significantly (*p* < 0.05) decreased MDA levels in the testis; and a substantial rise in the activities/levels of testicular SOD, GSH, and catalase in DOX-injected rats compared to rats that received only DOX (Table 1).

### 3.2. α-Bisabolol Activates Nrf2 Signaling and Triggers Upregulation of Antioxidant Defenses in DOX-Induced Testicular Injury in Rats

Rats injected with DOX exhibited significant (*p* < 0.05) downregulation in the expression levels of testicular proteins, Nrf2, SOD1, SOD2, catalase, and GPx1 and significant (*p* < 0.05) upregulation in the testicular protein expression levels of NOX2 and NOX4 compared to normal control rats. In addition, α-bisabolol-treated rats showed a significant (*p* < 0.05) increase in the testicular protein expression levels of Nrf2, SOD1, SOD2, catalase, and GPx1 and considerably (*p* < 0.05) decreased expression levels of NOX2 and NOX4 compared to normal control rats (Figure 2).

### 3.3. α-Bisabolol Attenuates the Levels and Expression Levels of Pro-Inflammatory Cytokines in DOX-Induced Testicular Injury in Rats

A significant (*p* < 0.05) rise in the levels and expression levels of TNF-α, IL-1 β, and IL-6 was observed in the testicular tissue of DOX-injected rats in comparison with normal control rats. Treatment with α-bisabolol considerably (*p* < 0.05) inhibited the DOX-induced rise in the levels and expression levels of testicular pro-inflammatory cytokines compared to treatment with DOX alone (Figure 3).

### 3.4. α-Bisabolol Protects the Testicular Architecture in DOX-Induced Testicular Injury in Rats

Testicular sections of normal and α-bisabolol-treated rats appear devoid of remarkable alterations in the testicular and epididymis architecture. However, histological examination of testicular sections in DOX-injected rats revealed severe derangement of testicular morphology. Treatment with α-bisabolol reinstates the near-normal testicular architecture in rats (Figure 4).

### 3.5. α-Bisabolol Attenuates the Expression Levels of Inflammatory Mediators and Downregulates NF-κB/MAPK Signaling Pathway in DOX-Induced Testicular Injury in Rats

The expression levels of iNOS, COX-2, p-NF-κB, p-IκB, and p-p38 in the testes of rats treated with DOX alone were significantly (*p* < 0.05) increased compared to those in normal control rats. α-Bisabolol treatment considerably (*p* < 0.05) reduced the increased testicular expression levels of iNOS, COX-2, p-NF-κB, p-IκB, and p-p38 compared to those in rats treated with DOX alone (Figure 5).

### 3.6. α-Bisabolol Attenuates ER-Stress-Mediated Testicular Apoptosis in DOX-Induced Testicular Injury in Rats

DOX injection caused a significant (*p* < 0.05) rise in the rats’ testicular protein expression levels of GRP-78, Bax, cleaved caspase-9, cleaved caspase-3, and cytochrome-C and significant (*p* < 0.05) decline in Bcl2 and Bcl-xL expression levels compared to those of normal control rats. Meanwhile, the administration of α-bisabolol to DOX-injected rats significantly (*p* < 0.05) downregulated the testicular protein expression levels of GRP-78, Bax, cleaved caspase-9, cleaved caspase-3, and cytochrome-C and significantly (*p* < 0.05) increased the expression levels of testicular Bcl2 and Bcl-xL in comparison with those of rats treated with DOX alone. The results have clearly revealed that α-bisabolol protects the testis by efficiently modulating ER-stress-mediated testicular apoptosis in DOX-injected rats (Figure 6).

## 4. Discussion

Testicular dysfunction associated with extensive gonadal damage is one of the serious side effects of cytotoxic cancer chemotherapy [31]. Among many chemotherapeutic agents, DOX, a well-known chemotherapeutic agent, has been associated with perturbed testicular functions and spermatogenesis [32]. Histopathological alterations and changes in testicular weight are some of the sensitive parameters used to detect toxicity in male reproductive systems. It is well documented that DOX exposure seriously affects testicular morphology, and it has a significant effect of testicular weight loss [33]. This is ascribed to spermatogenic damage in the testis and a considerable decrease in sperm count with parenchymal atrophy in the seminiferous tubules [34]. Histologically, DOX injections showed reduced size, irregularity, and a number of seminiferous tubules with reduced seminiferous epithelial layers [33]. Changes in the testicular architecture observed in our study might be explained by the direct and indirect effects of DOX, as DOX triggers lipid peroxidation and culminates in disrupting testicular structure and function [34]. α-Bisabolol has been reported to show negative effects in triggering reproductive or developmental toxicity in rats [35].

In recent years, significant attention has been given to natural products, mainly phytochemicals of plant origin which are widely distributed in edible plants and plant-based foods and are a source of novel molecules with health-promoting and medicinal benefits, particularly in improving the reproductive system. The occurrence of oxidative stress and inflammation complementing each other plays an important role in DOX-induced testicular injury, as reported previously [34]. DOX induces testicular oxidative stress, DNA damage, and apoptosis, which decrease sperm quality and motility and finally result in sexual dysfunction [36]. Additionally, the ring structure present in DOX enhances the non-enzymatic and enzymatic single-electron redox cycle associated with the generation of ROS from molecular oxygen [37]. DOX-induced free radical production depletes the antioxidant defense systems (GSH, catalase, and SOD), triggering the oxidation of proteins and lipids [38].

Additionally, NADPH oxidases are widely regarded as being a dominant source of ROS by regulating redox signaling [39]. NOX2 and NOX4 are the primary sources of ROS production in many tissues suffering the deleterious effects of oxidative stress [40]. In general, NOX enzymes are widely expressed in many organs, including the testis [41]. NOX2 and NOX4 are pro-oxidants, and they are structurally very active in accelerating the production of nitric oxide (NO) and superoxide radicals. Very high expression levels of these NADPH oxidases were reported in DOX-induced testicular damage in mice [42]. An enhanced generation of ROS has been associated with the DOX-induced cytotoxic effect on cancerous cells. In our study, improved activity and expression of SOD, catalase, and GSH along with a reduction in NOX2, NOX4, and NADPH oxidase following treatment with α-bisabolol demonstrate the attenuation of oxidative stress triggered by DOX.

One of the significant endogenous antioxidant defense systems that remove increased ROS accumulation involves the activation of the Nrf2 signaling pathway [43]. Nrf2 is a transcriptional factor that integrates the involvement of stress-mediated signaling at cellular levels. Nrf2 downregulation was reported to promote oxidative stress via increasing ROS-mediated lipid peroxidation and decreasing antioxidant enzyme activities [44]. DOX has been reported to cause perturbation of Nrf2, which regulates the expression of genes involved in anti-inflammatory and antioxidant processes and enzymes involved in fatty acid oxidation, mitochondrial respiration, and mitochondrial biogenesis. Mitochondria play an important role in the ROS-mediated apoptotic process, and DOX triggers mitochondrial impairment followed by an efflux of intracellular ROS. α-Bisabolol treatment revealed remarkable improvement in the antioxidant defense mechanisms, namely upregulation of Nrf2 and increased expression of antioxidant enzymes concomitant with a reduction in the lipid peroxidation product MDA in response to DOX-induced oxidative injury and associated sequelae. In addition, its impact in upregulating the testicular Nrf2 protein expression levels clearly revealed the ability of α-bisabolol in invoking Nrf2 signaling along with maintaining antioxidant defense systems which are responsible for the decreased oxidative stress parameters observed in our study.

In addition to oxidative stress, DOX has been well known to trigger the release of pro-inflammatory cytokines (IL-1β, IL-6, and TNF-α) and the induction of inflammatory enzymes (iNOS and COX). The findings of the present study also showed that DOX triggers inflammation, as evidenced by the induction of pro-inflammatory cytokines and enzymes with a concomitant loss in antioxidant defense enzymes and lipid peroxidation. However, α-bisabolol has been found to reduce the release of cytokines from the inflammatory cells, along with downregulating the inflammatory enzyme mediators and attenuating NF-κB, in agreement with previous numerous reports wherein DOX-induced testicular toxicity was attenuated by constituents of natural origin [45].

The transcriptional activator NF-κB regulates various inflammatory factors and plays a significant role in the pathophysiology of the DOX-induced inflammatory cascade in organ injuries [46,47,48]. NF-κB activation influenced by ROS overproduction is mainly responsible for the changes in the inflammatory cascades via the mediation of pro-inflammatory cytokines and mediators (TNF-α, IL-1β, IL-6, COX-2, and iNOS) [49]. NF-κB activation is inevitable during the prevailing dominance of pro-inflammatory mediators, and this positive feedback mechanism is predicted to augment pro-inflammatory signals which aggravate tissue injuries [50]. DOX dictates the activation and phosphorylation of NF-κB which is closely associated with IκK, which actively triggers IκB phosphorylation and degradation [51,52]. NF-κB phosphorylation is triggered by IκK and IκB activation under the environment of oxidative stress and inflammation which results in irreversible inflammatory assault [53].

MAPK signaling molecules are one of the major regulators of cellular proliferation, differentiation, inflammatory mediators, and apoptosis [54]. MAPK p38 kinases are particularly responsible for inflammatory stimulation induced by DOX injection [55]. MAPK signaling has also been implicated in the activation of the Nrf2 signaling pathway which is usually mediated by the phosphorylation of Nrf2 or through the nuclear translocation of Nrf2. Interestingly, treatment with α-bisabolol exhibited a robust resistance against DOX-induced testicular damage, as evidenced by reduced domination of pro-inflammatory cytokines, mediators, and NF-κB and MAPK signaling proteins resulting from the activation of Nrf2 defense systems, which revealed its potent antioxidant and anti-inflammatory effect.

Apoptotic cell death triggered by DOX involves mitochondrial dysfunction, as evidenced by the upregulation of the pro-apoptotic protein Bax, the downregulation of the antiapoptotic protein Bcl-2, the enhanced release of cytochrome-C into the cytosol, and the cleavage of caspase-3 and -9. MAPK has the potential to activate NF-κB, which further stimulates downstream genes, mainly those regulating pro-inflammatory responses, and results in pathological conditions. Ultimately, an excess amount of free radical production triggers cytokine production and activates MAPK and NF-κB transcription factors and apoptosis [13]. Upon activation, p38 MAPK induces cytosolic cytochrome-C release from the mitochondria, which further triggers the mitochondrial apoptotic pathway [56,57].

Additionally, ROS overproduction possesses a bidirectional relationship with oxidative and endoplasmic reticulum (ER) stress, which causes a buildup of unfolded proteins in the ER [16]. ER functions are seriously affected by various factors, including oxidative stress, Ca^2+^ leakage, iron imbalance, hypoxia, protein overload, and hypoxia, which triggers ER stress resulting in the accumulation of misfolded/unfolded proteins and finally apoptosis [58]. GRP-78, a crucial ER molecular chaperone, was considerably upregulated along with the mitochondrial pathway of apoptosis in the testicular tissue of DOX-treated rats, as reported in a previous study [10]. Secondly, p-IRE-1, a downstream target of IRE-1 signaling, was significantly increased, indicating IRE1 signaling activation by DOX. Lastly, JNK, which is a downstream target of the IRE-1 signaling pathway, was upregulated, which is similar to previous findings [59]. The p38 MAPK signaling pathway is linked with the activation of ER stress, as reported earlier [60,61,62]. In the event of ER stress, phosphorylation of p38 MAPK results in perturbations of cellular homeostasis [63]. In the present study, treatment with oral doses of α-bisabolol appears to inhibit p38 MAPK signaling and ER-stress-mediated apoptosis by downregulating the IRE-1-JNK signaling pathway.

The findings of the present study also displayed similar changes in the expression of apoptotic markers in the testes of DOX-induced rats. The occurrence of apoptosis in the testis can be ascribed to the free radical-dominated activation of NF-κB and MAPK signaling proteins directly associated with the onset and progression of DOX-induced testicular damage. α-Bisabolol effectively invokes Nrf2-mediated antioxidant defense mechanisms and defends the testis from the alarming apoptotic signals by reducing the crosstalk between ER stress/apoptotic signals and MAPK and NF-κB signaling proteins in DOX-triggered testicular toxicity in rats. The positive outcomes of molecular and biochemical parameters observed following treatment with α-bisabolol are reconfirmed with histopathological observations in DOX-induced testicular injury in rats. The study findings evidently reveal the antioxidant, anti-inflammatory, membrane-stabilizing, and anti-apoptotic potential of α-bisabolol.

## 5. Conclusions

In conclusion, it appears that α-bisabolol treatment exerts protection against DOX-induced testicular toxicity and that the underlying mechanism is attenuation of oxidative stress, inflammation, and apoptosis, which are mediated by its potent antioxidant, anti-inflammatory, antiapoptotic and membrane-stabilizing properties. These effects are ascribed to its ability to neutralize the free radicals generated during the metabolism of DOX and its advantageous modulation of the NF-κB and MAPK signaling pathways as well as the ER-stress-mediated mitochondrial pathway of apoptosis. Based on the observations of the present study, α-bisabolol could be suggested for use as an agent or adjuvant with the chemotherapeutic drugs to attenuate their deleterious effects on many organs, including the testis, during DOX chemotherapy and thus improve the morbidity in cancer survivors.

## Figures and Tables

**Figure 1 nutrients-14-04648-f001:**
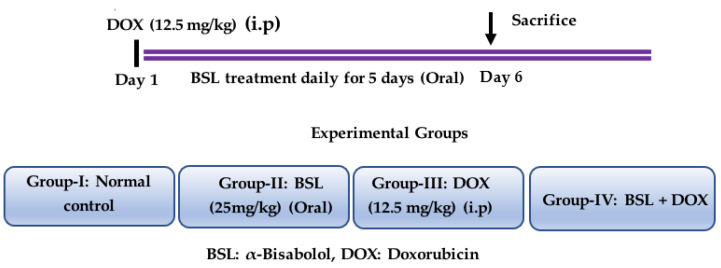
Study design and experimental groups.

**Figure 2 nutrients-14-04648-f002:**
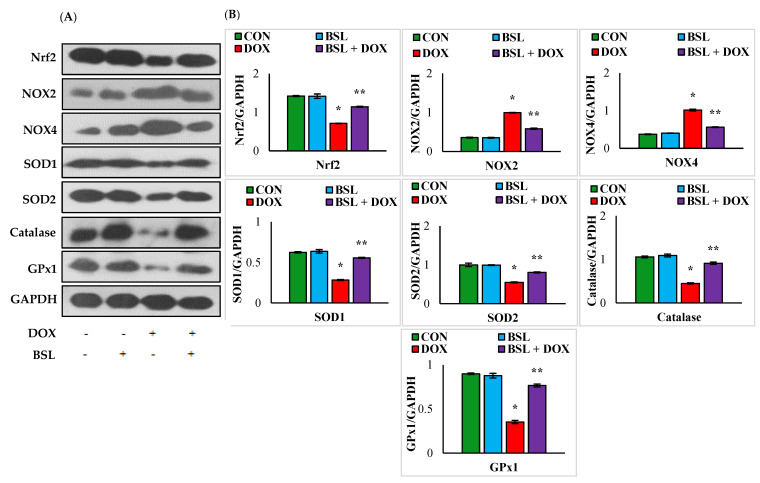
Effect of α-bisabolol on Nrf2 activation and expression levels of NADPH oxidases/antioxidants in DOX-injected rats. (**A**) Representative images of Western immunoblot analysis for Nrf2, NOX2, NOX4, SOD1, SOD2, catalase, and GPx1. (**B**) Densitometric analysis of testicular protein expression levels of Nrf2, NOX2, NOX4, SOD1, SOD2, catalase, and GPx1 assessed by Western blot analysis. Columns not sharing a common symbol (*, **) differ significantly from each other (* *p* < 0.05 vs. normal control, ** *p* < 0.05 vs. DOX control), CON-Control, BSL-α-Bisabolol, DOX-Doxorubicin, BSL + DOX-α-Bisabolol + Doxorubicin.

**Figure 3 nutrients-14-04648-f003:**
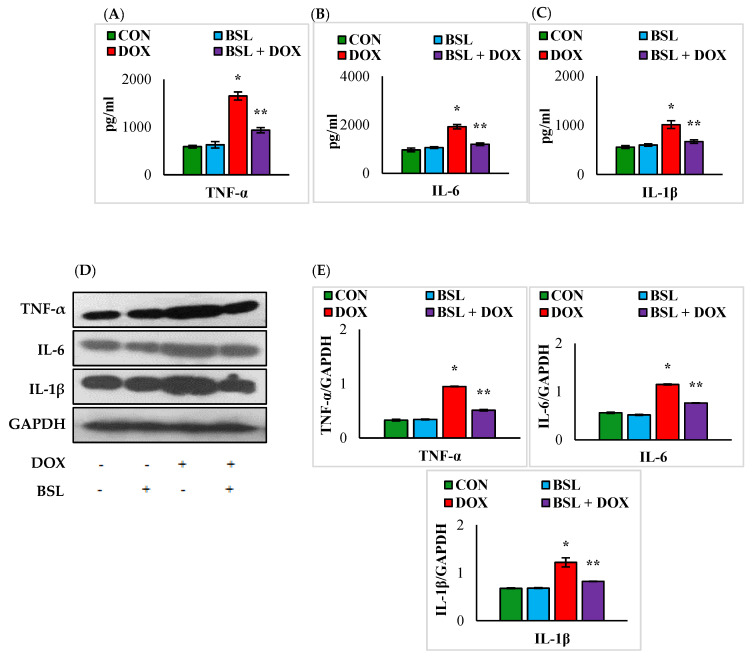
Effect of α-bisabolol on the levels/expression levels of pro-inflammatory cytokines. (**A**–**E**) Effect of α-bisabolol on the levels/expression levels of pro-inflammatory cytokines (TNF-α, IL-1β, and IL-6) in the testis. Each column is mean ± SEM for eight rats in each group; columns not sharing a common symbol (*, **) differ significantly from each other (* *p* < 0.05 vs. normal control, ** *p* < 0.05 vs. DOX control), CON-Control, BSL-α-Bisabolol, DOX-Doxorubicin, BSL + DOX-α-Bisabolol + Doxorubicin.

**Figure 4 nutrients-14-04648-f004:**
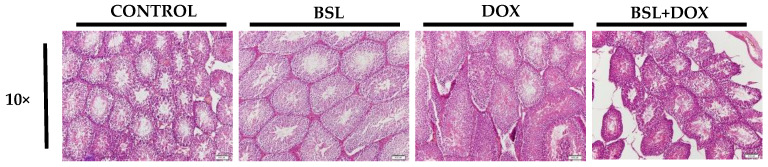
Histopathology of the testis. Testes of normal control rats and rats treated with α-bisabolol alone showed no pathological alterations. Rats treated with DOX alone showed severe derangement of testicular morphology, whereas α-bisabolol pre- and co-treatment reinstated the near normal testicular architecture in DOX-injected rats (10×), CON-Control, BSL-α-Bisabolol, DOX-Doxorubicin, BSL + DOX-α-Bisabolol + Doxorubicin.

**Figure 5 nutrients-14-04648-f005:**
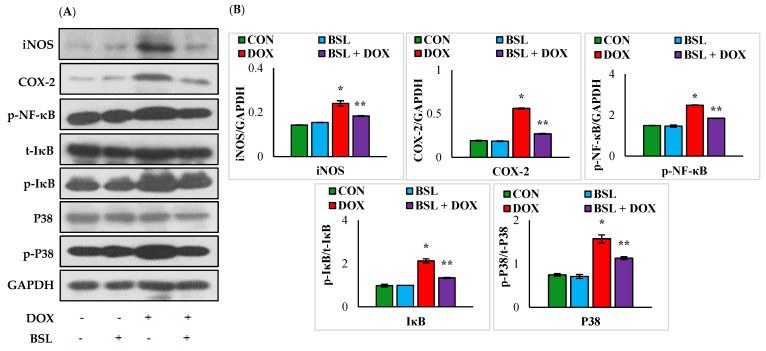
Effect of α-bisabolol on NF-κB/MAPK signaling in the testis. (**A**) Representative images of Western blot analysis for COX-2, iNOS, p-NF-κB-P65, p-IκBα, IκBα, p-p38, and p38. (**B**) Densitometric analysis of testicular protein expression levels of COX-2, iNOS, p-NF-κB-P65, p-IκBα, IκBα, p-p38, and p38 assessed by immunoblotting analysis. Columns not sharing a common symbol (*, **) differ significantly from each other (* *p* < 0.05 vs. normal control, ** *p* < 0.05 vs. DOX control), CON-Control, BSL-α-Bisabolol, DOX-Doxorubicin, BSL + DOX-α-Bisabolol + Doxorubicin.

**Figure 6 nutrients-14-04648-f006:**
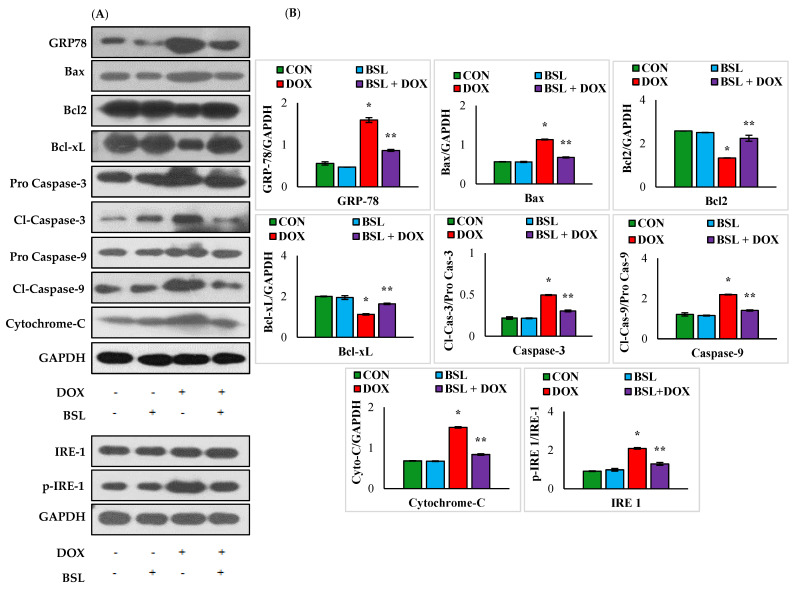
Effect of α-bisabolol on ER-stress-mediated intrinsic pathway of apoptosis in the testis. (**A**) Representative images of Western immunoblot analysis for Bax, Bcl2, Bcl-xL, procaspase-3, active caspase-3, procaspase-9, active caspase-9, cytochrome-C, IRE-1, and p-IRE-1 (**B**). Densitometric analysis of testicular protein expression levels of GRP-78, Bax, Bcl2, Bcl-xL, active caspase-3, active caspase-9, cytochrome-C, IRE-1, and p-IRE-1. Columns not sharing a common symbol (*,**) differ significantly from each other (* *p* < 0.05 vs. normal control, ** *p* < 0.05 vs. DOX control), CON-Control, BSL-α-Bisabolol, DOX-Doxorubicin, BSL + DOX-α-Bisabolol + Doxorubicin.

**Table 1 nutrients-14-04648-t001:** Effect of α-bisabolol on testicular weight, relative testicular weight ratio, and oxidative stress markers in DOX-injected rats.

Groups	Control	BSL	DOX	BSL + DOX
Testis weight (g)	3.62 ± 0.049	3.63 ± 0.041	2.48 ± 0.048 *	3.11 ± 0.078 **
Relative testis weight (%)	1.52 ± 0.017	1.50 ± 0.015	1.14 ± 0.030 *	1.35 ± 0.031 **
MDA (µM/mL)	56.17 ± 3.788	63.79 ± 1.31	103.53 ± 4.639 *	72.16 ± 4.532 **
SOD (U/mL)	36.61 ± 0.898	34.43 ± 1.386	19.02 ± 1.429 *	29.34 ± 1.946 **
Catalase (µM/min/mL)	84.46 ± 7.69	77.60 ± 12.927	28.14 ± 3.194 *	53.12 ± 3.648 **
GSH (µM/mL)	704.41 ± 36.847	684.36 ± 20.967	307.32 ± 15.269 *	580.59 ± 24.84 **

Each column represents mean ± SEM for eight rats in each group; columns not sharing a common symbol (*, **) differ significantly from each other (* *p* < 0.05 vs. normal control, ** *p* < 0.05 vs. DOX control), CON-Control, BSL-α-Bisabolol, DOX-Doxorubicin, BSL + DOX-α-Bisabolol + Doxorubicin.

## Data Availability

The data and articles referred to are incorporated and cited appropriately in the manuscript.

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
