# Peer review of "α-Bisabolol Attenuates NF-κB/MAPK Signaling Activation and ER-Stress-Mediated Apoptosis by Invoking Nrf2-Mediated Antioxidant Defense Systems against Doxorubicin-Induced Testicular Toxicity in Rats"

_nutrients, 2022, doi:10.3390/nu14214648_

Round 1

Reviewer 1 Report

Congratulation for the nice work. 

Minor typing and spelling errors occur, please check again the text.

Into the abstract for me seems to not be very clear that the doxorubicin is injected just one time. I suggest to reformulate some sentences.

The ER abbreviation is not explained at the first occurrence into the text (abstract). 

The point at the end of the sentences must be after the reference number and bracket.

For me are missing the numerical data. There are the figures, but I suggest to introduce also a table or more with exact numerical data.

At discussions there are not enough highlight the results and the conclusions regarding the studied substance. I suggest to reformulate, restructured the discussions.

Author Response

We sincerely thank the editor and reviewer for providing opportunity to revise the manuscript. We are grateful to the reviewer for constructive comments to improve the manuscript. As advised, all the suggestions have been incorporated in the revised manuscript.

Reviewer 2 Report

Comments for the authors:

Arunachalam et al. demonstrated that α-Bisabolol attenuates NF-κB/MAPK signaling activation and ER-stress mediated apoptosis by invoking Nrf2 mediated anti-oxidant defense systems against doxorubicin-induced testicular toxicity in rats. However, the current results are insufficient to support this conclusion. There are some issues should be addressed.

  1. Please add a graph with the in vivo experiments plan.
  2. The testicular functions other than testis weight should be determined, such as serum testosterone and ACP activity.
  3. p38 MAPK has been reported activating ER stress. However, ER stress also is known to activate the p38 MAPK pathway. The signal pathway should be clarified.
  4. The author described that ER-stress-mediated apoptosis was involved in doxorubicin-induced testicular toxicity. Which molecular mediates GRP78 downstream signaling? PERK, IRE1 or ATF6?
  5. To validate whether α-Bisabolol attenuates doxorubicin-induced testicular toxicity through invoking Nrf2. The knockdown experiments should be performed using cell line model.
  6. In Figure 4, α-Bisabolol treatment did not showed considerable restoration of testis histopathology in doxorubicin-treated rat. 
  7. Please add a graph with the proposed model.
  8. As shown in Figure 1A, it should be testicular weight rather than body weight.
  9. Please clearly indicate the treatment group in Figure 2,3,5,6.
  10. All Figures showed the paragraph markers, please correct.
  11. There some typos, such as “p-P38”. Please proofread thoroughly.

Author Response

(The authors gave the same response as above.)

Round 2

Reviewer 2 Report

Most of my concerns have been addressed.

Author Response

Dear Reviewer, Many thanks for your positive consideration of our research article. We are grateful for your positive and constructive comments on improving the manuscript. 

Dear Editor, The introduction and results, and Discussion are almost free from similarity. We sincerely thank you for considering our article and feel privileged for publishing in your reputed journal.

Many thanks and sincere regards